# Community-Based Healthcare for Migrants and Refugees: A Scoping Literature Review of Best Practices

**DOI:** 10.3390/healthcare8020115

**Published:** 2020-04-28

**Authors:** Elena Riza, Shona Kalkman, Alexandra Coritsidis, Sotirios Koubardas, Sofia Vassiliu, Despoina Lazarou, Panagiota Karnaki, Dina Zota, Maria Kantzanou, Theodora Psaltopoulou, Athena Linos

**Affiliations:** 1Department of Hygiene, Epidemiology and Medical Statistics, Medical School, National and Kapodistrian University of Athens, 115 27 Athens, Greece; 2Julius Center for Health Sciences and Primary Care, University Medical Center Utrecht, Utrecht University, 3584 CG Utrecht, The Netherlands; 3Renaissance School of Medicine, Stony Brook University, Stony Brook, NY 11794-8434, USA; 4Institute of Human Sciences, Wadham College, University of Oxford, Oxford OX1 3PN, UK; 5Prolepsis Institute for Preventive Medicine and Environmental and Occupational Health, 151 21 Marousi, Greece

**Keywords:** scoping review, migrants, refugees best practices, community-based healthcare

## Abstract

Background: Strengthening community-based healthcare is a valuable strategy to reduce health inequalities and improve the integration of migrants and refugees into local communities in the European Union. However, little is known about how to effectively develop and run community-based healthcare models for migrants and refugees. Aiming at identifying the most-promising best practices, we performed a scoping review of the international academic literature into effective community-based healthcare models and interventions for migrants and refugees as part of the Mig-HealthCare project. Methods: A systematic search in PubMed, EMBASE, and Scopus databases was conducted in March 2018 following the PRISMA methodology. Data extraction from eligible publications included information on general study characteristics, a brief description of the intervention/model, and reported outcomes in terms of effectiveness and challenges. Subsequently, we critically assessed the available evidence per type of healthcare service according to specific criteria to establish a shortlist of the most promising best practices. Results: In total, 118 academic publications were critically reviewed and categorized in the thematic areas of mental health (*n* = 53), general health services (*n* = 36), noncommunicable diseases (*n* = 13), primary healthcare (*n* = 9), and women’s maternal and child health (*n* = 7). Conclusion: A set of 15 of the most-promising best practices and tools in community-based healthcare for migrants and refugees were identified that include several intervention approaches per thematic category. The elements of good communication, the linguistic barriers and the cultural differences, played crucial roles in the effective application of the interventions. The close collaboration of the various stakeholders, the local communities, the migrant/refugee communities, and the partnerships is a key element in the successful implementation of primary healthcare provision.

## 1. Introduction

In recent years, over 2,000,000 migrants and refugees have fled civil unrest and socioeconomic instability and come to Europe since 2014 in search of a better and safer future [1]. A large portion of these individuals have come from the Middle East, fleeing conflicts such as the Syrian Civil War. Migrants and refugees worldwide often encounter substantial barriers to healthcare in their new home countries [2]. The World Health Organization in the 2018 “Report on the health of refugees and migrants in the European Union” [3] indicates that there are significant limitations to the accessibility and delivery of proper healthcare, as well as to the degree of effective communication. These limitations are mainly caused by differences in language, lack of knowledge regarding available services, limited participation in the economy, the healthcare system operability in each country, and the varying cultural attitudes and beliefs. As such, migrant and refugee populations are at higher risks of poverty and social exclusion. These barriers lead to inequitable access to healthcare, which is described as a fundamental human right. To reduce and prevent health inequalities among migrants and refugees in Europe, local healthcare systems will need to adapt to the specific needs of this population. There is evidence that integration into existing healthcare systems is promoted through tailored services at the level of local communities [4]. However, little is known on how to effectively develop and run community-based healthcare models for migrants and refugees.

In an effort to provide evidence-based information and practical guidance to the health professionals working at the primary healthcare level primarily in the EU Member States, the Mig-HealthCare project was launched in May 2017 (www.mighealthcare.eu) [5], partially funded by the European Commission Consumer, Health, Agriculture and Food Executive Agency (CHAFEA). The project’s aim is to produce a roadmap to effective community-based healthcare models in order to improve physical and mental healthcare services, to support the inclusion and participation of migrants and refugees in European communities, and to reduce health inequalities.

As part of the activities planned within this project, a systematic search in scientific databases was conducted with the objective to identify effective community-based healthcare models and interventions for migrants and refugees that could be used as best practices.

We performed a comprehensive review based on a search of the international, peer-reviewed literature to identify requirements, prerequisites, and concrete steps to design and implement community-based healthcare models serving migrants and refugees. The first step was to map the different models that are reported worldwide in the academic literature, along with their characteristics, core elements, and reported outcomes. Secondly, we critically analyzed the effectiveness of the models and interventions by applying prespecified criteria. Thirdly, based on our critical analysis and criteria evaluation, a shortlist of potential best practices and tools was created. Our findings are intended to provide policy-makers and health service providers working with migrant and refugee populations the concrete steps to successfully develop strategies to address and prevent health inequalities and foster integration at the level of local communities in Europe. 

One of the immediate challenges of this task was the controversy surrounding the terms community-based healthcare and community health. The terms are often used in different contexts, and different countries may use the terms in different ways. Nevertheless, we believe meaningful results can be obtained from a review of publications explicitly addressing community-based models and interventions. For the purpose of this review, we use a broad conception of community as “a group of inhabitants living in a somewhat localized area under the same general regulations and having common norms, values, and organizations” [6]. Community health refers to the health status of a defined group of people and the actions and conditions, both private and public (governmental), to promote, protect, and preserve their health [7]. Migrant and refugees are terms that are often used interchangeably, but they are defined by the United Nations High Commission for Refugees (UNHCR) as follows [8]: Migrants: “While there is no formal legal definition of an international migrant, most experts agree that an international migrant is someone who changes his or her country of usual residence, irrespective of the reason for migration or legal status. Generally, a distinction is made between short-term or temporary migration, covering movements with a duration between three and 12 months, and long-term or permanent migration, referring to a change of country of residence for a duration of one year or more”.

Refugees are “persons who are outside their country of origin for reasons of feared persecution, conflict, generalized violence, or other circumstances that have seriously disturbed public order and, as a result, require international protection. The refugee definition can be found in the 1951 Convention and regional refugee instruments, as well as UNHCR’s Statute”.

## 2. Materials and Methods 

### 2.1. Search Strategy and Selection

A literature search was performed for articles published in the English language following the PRISMA statement [9] in March 2018 in the databases: PubMed, EMBASE, and Scopus. Keywords and terms used were: “migrant”, “immigrant”, “refugee”, “asylum-seeker”, “healthcare”, “community-based”, and “model”, combined with an “AND” and/or an “OR”. 

No limits for publication dates were set; however, we divided our search in pre and post-2012 publication dates in an effort to effectively capture data on the recent migrant/refugee influx into Europe after 2011 following the civil unrest in countries of the Middle East and Africa, as our effort is of particular relevance to the present migrant/refugee crisis in Europe. Publications dated before 2012 have been published mostly with respect to migrant/refugee populations in countries outside Europe, such as the United States and Australia, in years predating the current European influx of migrants and refugees from Africa and the Middle East. 

#### Inclusion and Exclusion Criteria

Publications were eligible for review if they provided a comprehensive description of community-based models for healthcare delivery to migrant/refugee/asylum-seeking populations or other relevant minorities, as the provision of healthcare in some of these groups depends on legal status. In our selection of relevant publications, we employed a fairly broad concept of healthcare to deliberately cover different types of healthcare services and migrant/refugee population subgroups, such as adolescents, mothers, chronic patients, and migrants. This was done in order to produce a shortlist of potential best practices and tools covering the whole range of community-based health services so as to provide a broad base of information useful to policy-makers, researchers, and funders who work in this field. Eligibility was not restricted to models and interventions for specific groups of migrants and refugees. We included all ages, ethnicities, refugees, and migrants of any status. Explicit mentioning of the terms “vulnerability” or “vulnerable” was not required for inclusion, as we considered all migrants and refugees inherently vulnerable. All publications that proposed, discussed, or formally assessed a community-based model or intervention were included in the systematic collection and analysis. We excluded publications that only reported on health needs, barriers, and challenges to healthcare access among migrants and refugees without containing the element of specific practices and tools. Papers reporting on methods for participatory community-based health research, healthcare models strictly for rural or low-resource areas, and papers on models to engage migrants in clinical research were also excluded. Abstracts and conference proceedings were excluded from the formal analysis, as an in-depth critical review was not possible to the same extent as it was for full-text publications. 

### 2.2. Data Extraction

For all articles included in the final analysis, data was extracted on the following variables: (1) full citation, (2) year of publication, (3) type of study/paper, (4) country of implementation, (4) target population, (5) type of care/health needs, (6) model/intervention (keywords), (7) basic characteristics of the model/intervention, (8) best practices, (9) lessons learned, and (10) challenges and limitations. For the critical appraisal, we also extracted data on: (11) mode of evaluation (according to study design); (12) duration of follow-up (evaluation); (13) study sample (if applicable); and (14) theoretical underpinnings (e.g., for interventions based on behavioral or other models related to perceptions, attitudes, and barriers to change habits). All 3054 identified articles were screened by three independent reviewers, and the results were jointly discussed.

To facilitate the analysis, the publications were grouped by indication/disease area through an iterative process and, subsequently, categorized as models or interventions for health promotion/education, prevention, or disease management. 

### 2.3. Critical Assessment for Best Practices

We define “best practices” as interventions for which there is evidence to substantiate (at least some form of) effectiveness. To assess effectiveness, interventions described per category were evaluated based on the following criteria: study design; sample size; duration of follow-up; whether the study population was of Middle Eastern/North African descent (as this relates strongly to the increased influx of migrants/refugees into Europe following the civil unrest in several countries in the area, such as Syria, Iraq, Libya, and Lebanon); reported specific outcomes/advocate evidence-based approach; presence of theoretical underpinnings; and potential for reproducibility. For each of the above-mentioned variables, a marking scheme with subcategories was applied, and the total score was calculated for each practice to assess the level of evidence for the effectiveness of interventions and models for community-based healthcare for migrants and refugees (Table 1). Each study was assessed according to the criteria set in an Excel file, and the total score was computed automatically as the addition of the subscores in each category.

No score threshold was set, as this was a comparative process among the interventions identified in this review. The higher the total score, the more scientifically robust the proposed intervention was indicated.

## 3. Results

From our systematic database search, we retrieved 3054 unique records (Figure 1). Close screening of titles and abstracts narrowed the full text number down to 280 publications, including 22 abstracts and conference proceedings that were subsequently removed as they were not followed by any full-text publication. Based on the predefined inclusion and exclusion criteria, a total of 118 publications remained for data extraction. A full overview of the selection process is presented in Figure 1. 

### 3.1. Overall Study Characteristics

Out of 118 records, 53 (44.5%) discussed mental health, 36 (29.4%) community-based health services, 13 (10.9%) noncommunicable diseases (excluding mental health), 9 (7.6%) primary healthcare, and 7 (5.9%), maternal/women’s health and child health. 

Countries or regions of implementation included: North America (United States and Canada), 67/118 (56.7%), Europe, 28/118 (23.7%), Australia and New Zealand, 9/118 (7.6%), the Middle East, 6/118 (5.1%), Asia, 2/118 (1.7%), and Latin America, 1/118 (<1%). Five records did not specify the area of implementation. Populations targeted included migrants, immigrants, refugees, asylum-seekers, and racial and ethnic minorities, as defined by the respective authors. All these subgroups are part of the larger definition of migrants/refugees. Some publications targeted specific population groups such as women, children, adolescents, or families; elderly patients; trauma- or torture-exposed individuals; seasonal/farm workers; or individuals with a low income. Ethnicity or country of origin of the target population groups was mentioned in some, but not all, publications. Study designs included mostly mixed methods (use of quantitative and qualitative data); qualitative research (interviews and focus groups); surveys; and, less frequently, experimental designs (controlled, randomized, or pre-post-test designs). Most experimental studies were labeled as pilot studies. Community aspects were framed as either interventions implemented in the migrant community or as models or programs that rely on the engagement of different community stakeholders (such as universities, schools, and different community health services). The vast majority of studies (105 out of 118) were published between 2006 and 2018 (89%). A distinction between single and complex interventions was also made. By complex interventions, we mean activities (models or programs) that contain a number of component parts (interventions) with the potential for interactions between them that, when applied to the intended target population, produce a range of possible and variable outcomes [10]. Due to their nature, the effectiveness of complex interventions are more difficult to substantiate. 

### 3.2. Identified Best Practices According to Thematic Area of the Reviewed Records

#### 3.2.1. Mental Health

Published interventions on mental health were carried out primarily in the USA (25/53), Europe (12/53), and Canada (8/53) and less in the Middle East (2/53), Australia (2/53), and Asia (1/53). Dates of publication varied from 2002 to 2018, with 48/53 published in the period of 2005–2017. Target populations involved refugees (in some studies, further specified as of particular descent, children, tortured, newly arrived, families, and multi-ethnic adults); minorities (elderly, ethnic, low-income, and racial); immigrants (traumatized children and adolescents, women, low-income, and of specific/varying descent); asylum-seekers; and migrants (of specific descent, youth, children, and families).

Single interventions described for community-based mental health services pertained to the training of (future) healthcare workers and cultural brokering [11,12,13,14,15,16,17,18,19,20,21,22,23,24,25,26,27,28,29,30,31,32,33,34,35,36,37,38,39,40,41,42,43,44,45,46,47,48,49,50,51,52,53,54,55,56,57,58,59,60,61,62,63]. Training programs refer to the cross-cultural understanding and competency of healthcare workers [11,12] and the training and delivery of healthcare services among psychology or nursing students [14,15]. In terms of “cultural brokering” [12], community peers [17,18,19,20,21], bilingual gatekeepers [22], and ethnic matching of therapists and patients [13,24] were identified. Complex interventions constitute school-based programs to screen (and sometimes, also to treat) children and adolescents from migrant and refugee communities for mental health problems [16,25,26,27,28,29], mental health promotion in community day centers [30,31], and by community organizations [32,33] and various other community-based mental health services [30,31,32,33,34,35,36,37]. Screening tools for psychosocial risk assessments were also used [20,42,43].

Core elements of the identified interventions and models were: partnering with members from target communities [44,45]; community mobilization to stimulate outreach [33,46,47]; culturally and linguistically sensitive approaches [14,45,47,48,49,50,51,52,53]; education of health service providers on the needs of the target population [13,40,54]; awareness raising on mental health [46,55]; availability of information in relevant languages [44]; advocacy [56,57]; facilitating better integration [52]; responsiveness, coordination, and planning of different health and social services [12,54,55,58,59]; establishing a sense of belonging, community, and trust [18,58,59]; promoting empowerment and cultural competency [19,61,62]; funding [58]; and community-based participatory research [47].

#### 3.2.2. Health Services

Publications in this category are studies focusing on health promotion and access to care, mainly implemented in the USA (19/36), Europe (12/36), and Canada (2/36), and in Australia (2/36) and Asia (1/36), mostly in between 2007–2018 [64,65,66,67,68,69,70,71,72,73,74,75,76,77,78,79,80,81,82,83,84,85,86,87,88,89,90,91,92,93,94,95,96,97,98,99]. The populations addressed included racial and ethnic minorities; migrants (further specified as elderly, farm workers, or irregular); refugees (Syrian, apartment-dwelling, youth, older adult, and more); immigrants (elderly or recent); and minority children. It must be noted that the term “health services” in this review refers to all services related to health in general and not necessarily delivered within the healthcare system of which primary health care is an integral part.

Single interventions that emerged constituted of: providing health information [64,65], cultural brokering through ambassadors [66,67,68,69,70,71,72], bilingual advocacy and interpretation [73,74], and a community garden project addressing a sense of community and adoption of a healthy dietary pattern [75]. Complex interventions, on the other hand, concerned community-academic partnerships [76,77,78,79,80,81], community-based nursing initiatives [82,83,84], home-based health services [83], programs on prevention, and healthcare services for the uninsured [85].

Prevalent aspects of interventions were: supervision and responsibility of stakeholders to provide equity, cultural, and linguistic competence in healthcare access and delivery [86,87]; creating a sense of community and commitment; obligation of stakeholders [68,86,87,88]; community-based leadership that is transferring the operational supervision of the intervention to the local level to facilitate sustainability [79,89]; social networking viewed as a necessary skill along with good communication to improve the efficiency of the intervention [89,90]; and evidence-based guidelines [91,92].

Additionally, reducing discrimination [92]; the promotion of understanding of human values [84,93]; targeted outreach strategies with specific focus on health education, health promotion, disease screening, and prevention [94]; community collaboration and advocacy [74,88,91,93,94,95,96]; raising awareness on health risks [85,86,95]; and culturally and linguistically sensitive approaches [69,71,73,74,85,87,88,91,94,97,99].

Finally, building trust between migrants and service providers [77,81,84], educating service providers on the health needs of the community [77,79,81,85,90], warranting the availability of resources [85,97] and sustainability of the programs [79,91], surveillance, and the evaluation of interventions [79,88,92].

#### 3.2.3. Noncommunicable Diseases

Studies were conducted from 2012 to 2017 (12/13) in the USA (5/13), Europe (2/13), multiple countries (2/13), Canada, Australia, Middle East, and Latin America [100,101,102,103,104,105,106,107,108,109,110,111,112]. Target populations pertained to immigrants (mostly women of various descents), refugees, migrants, and ethnic minorities, some of which were further defined as diabetic.

Community-based strategies for the management of the following were discussed: cancer screening [100,101,102,103,104], diabetes mellitus [99,105,106,107,108], cardiovascular disease prevention [109,110], and other chronic diseases [111,112]. Cancer—mostly breast cancer—prevention tools involved culturally tailored, narrative educational videos [100], pictograph-enhanced instructions [102], and patient-centered strategies [103,104]. The latter was also applied in diabetes mellitus management [106]. Diabetes mellitus and cardiovascular disease prevention interventions were culturally tailored approaches and story-telling [98,103,104,105,107,108,109].

Core elements of the described practices are culturally and linguistically sensitive education [103,104,109,112], involvement and support of the migrant communities’ infrastructures [110], awareness-raising about health risks [101], outreach approaches through families and community peers [101,105,111], facilitating the “community voice”, intersectional collaboration, and funding [101].

#### 3.2.4. Primary Healthcare 

Publications in this area of action were conducted mainly between 2012 and 2018 (7/9) in Australia (2/9), the USA (2/9), Canada (2/9), and the Middle East (1/9) (+2/9 not specified), and populations addressed included refugees and asylum-seekers, immigrants of various descents, and vulnerable migrants [113,114,115,116,117,118,119,120,121]. Aspects of the interventions in primary healthcare discussed are engagement with the migrant community through partnerships [113,114], stronger focus on ancillary services [121], interdisciplinary collaboration between public health and primary care institutions [116,117,118], culturally and linguistically trained interpreters [118,119], evidence-based guidelines [118,119], outreach activities by nurses [120], training of staff in cultural competency [114,121], health promotion education among migrants, and advocacy [117,121].

#### 3.2.5. Maternal, Women’s and Child Health

Six of the seven identified records were carried out from 2001 to 2015 (1/7 in 1989) in the USA (3/7), Australia (2/7), Canada, and the Middle East (1/7). Target populations involved refugees (in certain studies, further defined as Syrian, children, and parents); immigrants; racial; and ethnic minority women [122,123,124,125,126,127,128].

The main focus of interventions in this area was the reduction of maternal and child health inequalities among migrant/refugee communities, mostly through publicly funded universal health activities [122] and government-led approaches [123]. Committed community and health service provider (agencies) partnerships through multiple mobilization strategies were considered successful for improving the health of pregnant women [124]. Capacity-building, as in ways to address barriers in healthcare provisions for minority populations, such as health insurance availability, healthcare cost reimbursement, healthcare advice in a native language, and culturally sensitive training of healthcare professionals, is essential; to maintain the interests of service providers and community members is essential for program sustainability. In general, partnerships between the target community and the different local healthcare providers are recommended to identify the barriers faced by women and potential solutions for improving access to care [125]. Intensive child health promotion and education campaigns using ethnic media (radio, TV channels, and newsletters in the native language of the beneficiaries) and social networking were observed to positively affect parental awareness, knowledge, and beliefs about infectious disease prevention in children [126].

For individuals with additional health needs, such as those requiring prenatal or pediatric care, a Culturally Appropriate Resources and Education (C.A.R.E.) Clinic Health Advisor is recommended for specialty clinics. This type of health advisor facilitates communication, establishes a sense of community, and helps patients navigate the healthcare system [127]. To implement reproductive health services in humanitarian emergencies, facilitators are a pre-existing functioning health infrastructure, with prior training in their particular type of service delivery, dedicated leadership, and the availability of sufficient funding and resources [128]. 

### 3.3. Promising Best Practices at the Community Level

Our assessment prioritized the 15 top best practices according to the set criteria. The top scores were 20 points (1 publication), 19 points (1), 18 points (1), 17 points (4), 16 points (1), and 15 points (7 publications). These 15 interventions best fit the set evaluation criteria and are presented as the most promising. In terms of the area of action, they are categorized as follows: seven in mental health, two in health service provision, two in noncommunicable diseases, two in primary healthcare, and two in maternal health (Table 2). 

The training of health professionals, close collaborations of stakeholders, partnerships, social networks, linguistically and culturally sensitive service provisions, participatory approaches, and advocacy are elements described in these promising best practices.

## 4. Discussion

This study reviewed the academic literature for best practices in community healthcare models for migrants and refugees.

We developed an evaluation tool to assess and classify the search results for their scientific robustness based on their reported characteristics, such as population size, type of intervention, achieved outcome, reproducibility, and theoretical underpinning.

In the final set of identified practices, five areas of action were identified: mental health, health services, noncommunicable diseases, primary healthcare, and maternal women’s and child health. All publications were thoroughly assessed per category, and then, based on the evaluation ranking (a numerical score), the identified interventions/best practices were prioritized in terms of scientific soundness and reproducibility potential. Partnerships between governments and community providers, the design and delivery of tailor-made educational activities for children, linguistically and culturally adapted disease prevention activities, and community and school-based interventions for mental health addressing various population groups, as well as training programs for future healthcare professionals have been shown to be efficient and reproducible ways to improve the health of vulnerable population groups such as refugees and migrants.

### Challenges-Limitations

This review is a comprehensive effort to identify community-based best practices at the primary healthcare level, addressing refugees and migrants in the peer-reviewed literature with the aim to provide information and guidance to the health professionals working at the primary healthcare level primarily in the EU Member States. This effort encountered several challenges/limitations. There is an abundance of publications regarding interventions for migrant/refugee healthcare in the peer-reviewed literature. A huge variation in the meaning of the terms community, community health or healthcare, and best practice was identified, along with an interchangeable use of the terms migrants and refugees, as well as immigrants, minorities, and asylum-seekers.The majority of publications (64.3%) originated from the US, Canada, and Australia, addressing, by large, refugees and migrants at a much-progressed social integration stage compared to Europe and from very different ethnic backgrounds.Many publications did not specify ethnicity; country of origin; or specific characteristics (i.e., age or social determinants) of the target population.Despite the richness of published information, it should be noted that multiple other interventions exist that have not (and may never) been published through a peer-review process, due to numerous reasons spanning from lower prioritization of the health issue to lack of resources to cover publications fees. Certainly, there can be areas of migrant/refugee health that could not be retrieved in the literature prior to March 2018, as no relevant publications were available. However, after reviewing some abstracts and conference proceedings, we have strong reasons to believe that many interventions delivered as pilot studies have not been published as full-text papers, despite the fact that they provide valuable insights into potentially effective community-based interventions.

Evidently, such issues are not of lesser importance compared to the published ones. In this aspect, it is important to note that studies on migrant/refugee health issues may never materialize into a peer-reviewed publication, as they often face several barriers such as the scarcity of systematically recorded data on migrant/refugee health and a reluctance of vulnerable populations to participate in interventions stemming from communication difficulties to legal issues of residence and social exclusion, resulting in small participation rates and study samples or a difficulty in monitoring health in populations on the move, as they often change locations or even countries. 

The objective of the present review was to identify the best practices and tools of community-based interventions for migrants/refugees, and as such, a set of 15 practices addressing the areas of mental health, primary healthcare, health service provision, and noncommunicable disease management and prevention strategies, as well as maternal and child health, were identified based on specific evaluation criteria.

The majority of projects, activities, and interventions identified in this review focus on the area of mental health, and this is an important finding that needs to be examined further, as there could be a multitude of reasons for this. The area of health service provision is also important, as well as the issue of chronic disease management, which poses as a major future challenge for healthcare systems. The primary healthcare setting is vital, as it has close links to the community and facilitates the involvement of the local population in preventing and managing diseases. It is important to note that, in almost all of the sources identified, the elements of good communication, the linguistic barriers, and the cultural elements played crucial roles in the effective applications of the interventions. Evidently, the close collaborations of the various stakeholders, the local communities, the migrant/refugee communities, and the partnerships are key elements in the successful implementation of effective primary healthcare provisions.

## 5. Conclusions

The provision of essential health services of good quality for all population groups of a society is described in the 2030 Agenda for Sustainable Development [129]. These services, as emerged from our scoping review, include health promotion, disease prevention, and disease-management activities and should aim to meet the needs of all people, especially migrants, refugees, and the vulnerable. Primary healthcare services offered at the community level can cover all aspects of health-related needs and are very effective in addressing the health needs and challenges of all.

## Figures and Tables

**Figure 1 healthcare-08-00115-f001:**
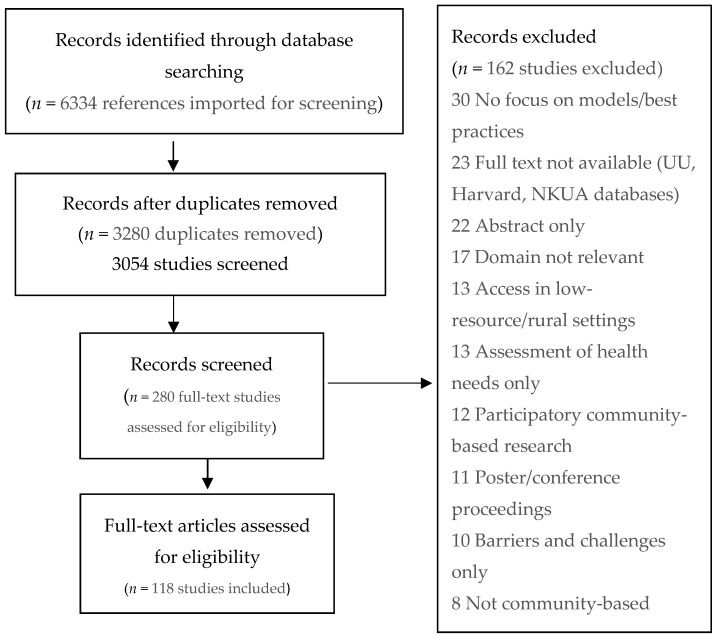
PRISMA flow diagram for Mig-HealthCare systematic database search.

**Table 1 healthcare-08-00115-t001:** Evaluation criteria of selected interventions on community-based best practices.

Study Design	Sample Size (*n*)	Duration of Follow-Up	Middle Eastern/North African Individuals Included in Target Population	Reported Outcomes/Or Advocate Evidence-Backed Approach	Reproducible (As Mentioned in Publication)	Theoretical Underpinning
0 = not specified	NA = not applicable	NA = not applicable	1 = no	1 = no	1 = not mentioned	NA = not applicable
1 = review/description (no data)	1 = <10	C = cross-sectional design	2 = yes	2 = yes	2 = can be reproduced	1 = not present in publication
2 = qualitative or quantitative data	2 = 11–50	1 = days				2 = presented in publication
3 = mixed methods	3 = 51–100	2 = weeks				
4 = experimental study (randomized, controlled, or pre-post-test design)	4 = > 100 or ≤ 10 papers (for reviews)	3 = months				
5 = literature review	5 = > 1000 or > 10 papers (for reviews)	4 = 1–5 years				
P = pilot study		5 = > 5 years				

**Table 2 healthcare-08-00115-t002:** The 15 highly assessed best practices.

Publication	Reference Number	Area of Intervention	Intervention	Score
McMurray (2014)	[116]	Primary healthcare	Partnership between a dedicated health clinic for government-assisted refugees, a local reception center, and community providers	20
Reavy (2012)	[127]	Maternal health	New clinic model for prenatal and pediatric refugee patients (in particular, the role of the Culturally Appropriate Resources and Education (C.A.R.E.) Clinic Health Advisor)	19
Small (2016)	[38]	Mental health	Comparison of three different treatment modalities: treatment as usual (TAU), home-based counseling (HBC), and a community-based psycho-educational group (CPG)	18
Bader (2006)	[109]	Noncommunicable Diseases	Linguistically and culturally sensitive cardiovascular disease (CVD) prevention program	17
Sheikh & McIntyre (2002)	[126]	Maternal health	Intensive child health promotion and education campaign using ethnic media and social network	17
Williams & Thompson (2011)	[40]	Mental health	Community-based mental healthcare services	17
Kaltman (2011)	[37]	Mental health	Collaborative mental healthcare program implemented in a network of primary care clinics that serve the uninsured	17
Fondacarro (2016)	[14]	Mental health	Training program for psychology students (“Connecting Cultures”)	16
Levin-Zamir (2011)	[114]	Primary healthcare	Cross-cultural program for promoting communication and health	15
Siddaiah (2014)	[112]	Noncommunicable Diseases	Community-based, culturally competent respiratory health screening and education	15
Tumiel-Behalter (2011)	[89]	Health service provision	Community program with a participatory approach to improve the health of four underserved communities (“Good For The Neighborhood”)	15
Ferrera (2017)	[96]	Health service provision	Health promotion initiative that integrates principles of positive minority youth development	15
Tyrer & Fazel (2014)	[29]	Mental health	School and community-based interventions	15
Kaltman (2016)	[36]	Mental health	Mental health intervention for primary care clinics that serve the uninsured	15
Goodkind (2014)	[56]	Mental health	Community-based advocacy and learning intervention with refugees and undergraduate students	15

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
