# Peer review of "Community-Based Healthcare for Migrants and Refugees: A Scoping Literature Review of Best Practices"

_healthcare, 2020, doi:10.3390/healthcare8020115_

Round 1

Reviewer 1 Report

I acknowledge the team's collaboration and efforts on this immense piece of work. There are a few findings but none are new and overall the work provides at best a mapping of approaches, and at worlds mistakes in their evaluation of effectiveness. 

I had several methodological concerns:

Phase 1 begins with mapping, of studies, authors maybe should have stopped here. Methods at this point were ok.  But there were some concerns re strange to see to two sets of searches, not sure how these were handled. pre and post 2012.

Phase 2-  critical appraisal using validated tools, is fine, but the authors have developed their own tool, and I am very concerned with the lack of validation but link to scores.

Phase 3- I feel there authors have grossly departed from a scoping method and moved into a realist approach bases on their many diverse slides. The problem is there is no RAMSES checklist or justification for this approach, and suggestion of effectiveness when robust systematic review methods exist, even for EU/EA, is concerning. Primary care, just as most healthcare resources are limited, and there needs to be appropriate methods to determine where to spend resources. 

Essentially authors had bypassed the standards associated with a scoping reviews. They have created criteria and analysis that are not methodologically sound in the context of a scoping review. Mapping is find, but the rest lasts methods. This approach is would need consensus methods, and they also have brought realist type methods.

I also am concerned they have also excluded participatory models, when community participatory approaches are one of the recognized models for vulnerable population. 

Author Response

Dear Reviewer, many kind thanks for your insightful comments which we have looked at very carefully. Please find below our responses. We hope that you will view them favourably. B.

Suggestions for Authors
Reviewer 1

I acknowledge the team's collaboration and efforts on this immense piece of work. There are a few findings but none are new and overall the work provides at best a mapping of approaches, and at worlds mistakes in their evaluation of effectiveness. I had several methodological concerns: Phase 1 begins with mapping, of studies, authors maybe should have stopped here. Methods at this point were ok. But there were some concerns re strange to see to two sets of searches, not sure how these were handled. pre and post 2012.

Authors’ Comment: It must be clarified here, that the search was not handled in any way differently before and after 2012, the division was to facilitate the organisation of the search and to put the publications retrieved into context. The year 2012 is the year after the escalation of the civil war in Syria following the riots of the Arab spring in the Middle East, therefore the composition of the refugee/migrant populations has changed. A clarification has been made in Lines 113,114.

Phase 2- critical appraisal using validated tools, is fine, but the authors have developed their own tool, and I am very concerned with the lack of validation but link to scores.

Authors’ Comment: For the critical appraisal we used validated guidance to identify best practices from the EU, which “can be used in any future action co-funded under the 3rd Health Programme to select best practices. Such actions would be free to decide whether these criteria are a guidance for best practice selection or if they would further adapt and develop e.g. into a specific evaluation matrix, depending on the topic of the action”. https://ec.europa.eu/health/sites/health/files/major_chronic_diseases/docs/sgpp_bestpracticescriteria_en.pdf. As this review forms part of a European project, we followed the relevant guidance. As for the scores, we used the scores to rank the publications according the set criteria to provide a quantitative estimate of adherence to the set criteria.

Phase 3- I feel there authors have grossly departed from a scoping method and moved into a realist approach bases on their many diverse slides. The problem is there is no RAMSES checklist or justification for this approach, and suggestion of effectiveness when robust systematic review methods exist, even for EU/EA, is concerning. Primary care, just as most healthcare resources are limited, and there needs to be appropriate methods to determine where to spend resources.

Essentially authors had bypassed the standards associated with a scoping reviews. They have created criteria and analysis that are not methodologically sound in the context of a scoping review. Mapping is find, but the rest lasts methods. This approach is would need consensus methods, and they also have brought realist type methods.

Authors’ Comment: These are all very useful comments for which we are thankful. Our aim for this review as part of a larger EU project was to identify best practices for community healthcare models for refugees and migrants in the peer reviewed literature, in order to identify key elements that can facilitate planning of healthcare delivery at the primary healthcare level in Europe, which is an indication for scoping review. For “best practice” we followed the EU proposed definition according to which we looked for process and outcomes with a clear context definition, along with elements of transferability, sustainability and participation of stakeholders. We grouped the findings according to the area of intervention and we ranked them according to the best practice criteria in order to identify the most effective ones. Our results can inform model practices for refugees and migrants at the European level. We understand realist synthesis as an approach to provide insight into social policies and complex interventions that cannot be measured directly. One of the items of the RAMESES publication standards for realist syntheses is “Scoping the literature” (Wong et al. BMC Medicine 2013, 11:21 http://www.biomedcentral.com/1741-7015/11/21) which is our effort to present the existing information in community healthcare interventions for migrants/migrants.

I also am concerned they have also excluded participatory models, when community participatory approaches are one of the recognized models for vulnerable population.

Authors’ Comment: We must stress the fact that this review forms one approach to identify best practices within the Mig-Health Care project and that participatory research was foreseen and was conducted as a separate study which will be presented in a separate publication.

Reviewer 2 Report

This scoping review on best practices and interventions comes at a good time for migrant and refugee health. Most studies past and present focus largely on the challenges and barriers that migrants and refugees experience pre-and post-migration. While these studies remain critical, we must begin to shift our thinking towards interventions that work towards the inclusion of migrants and refugees in the health care systems of their host countries. 

In line 46 and 47, there is reference to large numbers of migrants and refugees, without giving the numbers. The UNHCR has this information available on their website and it may give a better picture as to the extent of migration by adding the numbers. 

In lines 51, 53, 54, 55, there are no in-text citations/ references for the statements made. Also, there is reference made to poverty (line 54) as a consequence of the limitations mentioned in lines 51, 52, and 53. But these limitations do not mention for example, limited participation in the economy (which is an exclusionary practice). 

The authors place emphasis on the term vulnerable - we know from the literature that by virtue of being a migrant or refugee, one is already vulnerable, so I'm not sure that use of the term gives any kind of conceptual weight to the plight of migrants and refugees. For example, what are the parameters for interventions for vulnerable migrants and refugees specifically [line 70]. So is there a difference between those who aren't vulnerable and in the context of health care, what does this actually mean? In line 113, 114, the authors are explicit about considerations about all migrants and refugees as vulnerable. It is a contradiction to include the word vulnerable in the title and the objective and should be removed altogether. 

Line 89/90: Expand on the statement of migrant and refugee definitions as it relates to the scoping review. I realise that the terminology can be challenging because there is no universally accepted definition. For example, the IOM uses an umbrella term of migrant, of which refugees are a sub set. This is ok but becomes challenges when one starts to look at entitlements and benefits (which are usually linked to migrant status). 

I think it is important to be consistent with the presentation of the term migrant and refugee throughout the paper. In lines 98, 99, 100, there is reference made to refugee/migrant, then in line 102, reference is made to migrants and refugees. Does the differences in the ways these terms are presented give different meanings to the concepts of migrants and refugees? Later still, asylum seekers are introduced, but the writing only delineates between migrants and refugees. I think it is important to clarify this to the reader or to be consistent in the way these terms are presented. 

Line 131: "Articles were screened..." how many articles? All articles or a sub set?

Line 140: There is reference made to whether the study population was of Middle Eastern/ Northern African descent - why was this part of the assessment when there was no justification for this. While there is mention of a large portion of individuals have come from the Middle East (line 48), there are no statistics that back this up nor motivate for this as a critical criteria. 

Line 145: The ranking or scores assigned are not made clear from this section nor the table (Table 1). What is a high score, what is a low score? How was this computed? Manually? Excel? Application-aided programme?

Line 172: The different migrant terminologies should be outlined upfront. Please reconfirm that racial minorities are indeed part of the migrant discourse. 

Lines 229-238: Expand on the prevalent aspects of interventions so they make more sense to the reader. For example, obligation of stakeholders - to who and what are they obligated to; what about community-based leadership; what about social networking; targeted outreach strategies (such as)?

Line 274: "agency partnerships" - who are these agencies?

Line 275: "Capacity building to maintain the interest..." - what does this mean in this context of capacity building?

Line 279: "...ethnic media..." what is ethnic media?

Line 285: "...navigate the healthcare system...(REFERENCE MISSING)

Line 302: see my previous comment on vulnerable

Line 310: "Maybe quickly summarise findings here" = remove

Line 354: Please cite the UN SDGs as you make reference to it

Line 356: Emphasise migrants and refugees in your conclusion

Author Response

Dear Reviewer,

many kind thanks for your valuable and insightful comment, which we have considered very carefully.

Please find below our responses. We hoe that you will view them favourably.

Comments and Suggestions for Authors
- Reviewer 2

This scoping review on best practices and interventions comes at a good time for migrant and refugee health. Most studies past and present focus largely on the challenges and barriers that migrants and refugees experience pre-and post-migration. While these studies remain critical, we must begin to shift our thinking towards interventions that work towards the inclusion of migrants and refugees in the health care systems of their host countries. 

In line 46 and 47, there is reference to large numbers of migrants and refugees, without giving the numbers. The UNHCR has this information available on their website and it may give a better picture as to the extent of migration by adding the numbers. 

Authors’ Comment: A clarification was added in Lines 46-48.

 “In recent years, large numbers of migrants and over 2,000,000 migrants and refugees have fled civil unrest and socioeconomic instability and come to Europe since 2014 in search of a better and safer future (https://data2.unhcr.org/en/situations/mediterranean)”.

In lines 51, 53, 54, 55, there are no in-text citations/ references for the statements made. Also, there is reference made to poverty (line 54) as a consequence of the limitations mentioned in lines 51, 52, and 53. But these limitations do not mention for example, limited participation in the economy (which is an exclusionary practice). 

Authors’ Comment: A clarification has been made in lines 51-55.

“The World Health Organisation in the 2018 “Report on the health of refugees and migrants in the European Union” (www.euro.who.int) indicates that there are significant limitations to the accessibility and delivery of proper healthcare as well as to the degree of effective communication. These limitations are mainly caused by differences in language, lack of knowledge regarding available services, limited participation in the economy, the healthcare system operability in each country, and the varying cultural attitudes and beliefs”.

The authors place emphasis on the term vulnerable - we know from the literature that by virtue of being a migrant or refugee, one is already vulnerable, so I'm not sure that use of the term gives any kind of conceptual weight to the plight of migrants and refugees. For example, what are the parameters for interventions for vulnerable migrants and refugees specifically [line 70]. So is there a difference between those who aren't vulnerable and in the context of health care, what does this actually mean? In line 113, 114, the authors are explicit about considerations about all migrants and refugees as vulnerable. It is a contradiction to include the word vulnerable in the title and the objective and should be removed altogether. 

Authors’ Comment: This review forms a component of a European project entitled “Strengthen Community Based Care to minimize health inequalities and improve the integration of vulnerable migrants and refugees into local communities” that is why we introduce the term “vulnerable”.

By the term vulnerable migrants and refugees we refer to sub-groups such as children, pregnant and breast-feeding mothers and those with chronic diseases that have an additional element of health-related vulnerability. However, as this term indeed causes some confusion, we removed the term “vulnerable” from the title and the text (lines 73, 313, 333, 382).

Line 89/90: Expand on the statement of migrant and refugee definitions as it relates to the scoping review. I realise that the terminology can be challenging because there is no universally accepted definition. For example, the IOM uses an umbrella term of migrant, of which refugees are a sub set. This is ok but becomes challenges when one starts to look at entitlements and benefits (which are usually linked to migrant status). 

Authors’ Comment: Migrant and refugee definitions have been added (lines 93-104)

I think it is important to be consistent with the presentation of the term migrant and refugee throughout the paper. In lines 98, 99, 100, there is reference made to refugee/migrant, then in line 102, reference is made to migrants and refugees. Does the differences in the ways these terms are presented give different meanings to the concepts of migrants and refugees? Later still, asylum seekers are introduced, but the writing only delineates between migrants and refugees. I think it is important to clarify this to the reader or to be consistent in the way these terms are presented. 

Authors’ Comment: The reference is presented as migrants/refugees throughout the text (Lines 113,114,116, 121, 348, 350, 368).. The reference to asylum seekers has been made as in several countries the provision of health services is linked to asylum status, hereby differentiating them from the general group of migrants/refugees. A clarification has been introduced in the text (lines 122,123).

Line 131: "Articles were screened..." how many articles? All articles or a sub set?

Authors’ Comment: All 3,054 identified articles were screened (clarification made in line 148).

Line 140: There is reference made to whether the study population was of Middle Eastern/ Northern African descent - why was this part of the assessment when there was no justification for this. While there is mention of a large portion of individuals have come from the Middle East (line 48), there are no statistics that back this up nor motivate for this as a critical criteria. 

Authors’ Comment: The vast majority of migrant/refugee influx into Europe since 2011 comes from countries of the Middle East and Northern Africa e.g. Syria, Iran, Iraq, Lebanon, Libya as also described by the UNHCR statistics.

A clarification has been added in lines 157-159:

“To assess effectiveness, interventions described per category were evaluated based on the following criteria: study design, sample size, duration of follow-up, whether the study population was of Middle Eastern/North African descent (as this relates strongly to the increased influx of migrants/refugees into Europe following the civil unrest in several countries in the area such as Syria, Iraq, Libya, Lebanon), reported specific outcomes/advocate evidence-based approach, presence of theoretical underpinnings and potential for reproducibility”.

Line 145: The ranking or scores assigned are not made clear from this section nor the table (Table 1). What is a high score, what is a low score? How was this computed? Manually? Excel? Application-aided programme?

Authors’ Comment: Each study was assessed according to the criteria set in an excel file and the total score was computed automatically as the addition of the sub-scores in each category. No score threshold is set as this is a comparative process among the interventions identified in this review.

A clarification is added in Lines164-167.

“Each study was assessed according to the criteria set in an excel file and the total score was computed automatically as the addition of the sub-scores in each category”.

Line 172: The different migrant terminologies should be outlined upfront. Please reconfirm that racial minorities are indeed part of the migrant discourse. 

Authors’ Comment: These are the terminologies used by the authors of the identified publications and all these sub-groups are part of the larger term migrants/refugees.

 A clarification has been added in Lines195-196.

“All these sub-groups are part of the larger term migrants/refugees”.

Lines 229-238: Expand on the prevalent aspects of interventions so they make more sense to the reader. For example, obligation of stakeholders - to who and what are they obligated to; what about community-based leadership; what about social networking; targeted outreach strategies (such as)?

Authors’ Comment: A clarification has been made in Lines 252-261

“Prevalent aspects of interventions were: supervision and responsibility of stakeholders to provide equity, cultural and linguistic competence in healthcare access and delivery, [82, 83], creating a sense of community and commitment, obligation of stakeholders [63, 82-84], community-based leadership, that is transferring the operational supervision of the intervention to the local level to facilitate sustainability [74, 85], social networking, viewed as a necessary skill along with good communication to improve efficient of the intervention [85, 86], evidence-based guidelines [87, 88].

Reducing discrimination [88], promotion of understanding of human values [79, 89], targeted outreach strategies with specific focus on health education, health promotion, disease screening and prevention [90], community collaboration and advocacy [68, 84, 87, 89-92], raising awareness on health risks [81, 82, 91], culturally and linguistically sensitive approaches [64, 66, 68, 69, 81, 83, 84, 87, 90, 93]”.

Line 274: "agency partnerships" - who are these agencies?

Authors’ Comment: A clarification has been made in Line 301.

“Committed community and health service providers (agencies) partnerships through multiple mobilization strategies were considered successful for improving the health of pregnant women”.

 Line 275: "Capacity building to maintain the interest..." - what does this mean in this context of capacity building?

Authors’ Comment: A clarification has been made in Lines 303-305.

“Capacity building, as in ways to address barriers in healthcare provision for minority populations, such as health insurance availability, healthcare cost reimbursement, healthcare advice in native language, culturally sensitive training of healthcare professionals) is essential to maintain the interest of service providers and community members was essential for programme sustainability”.

Line 279: "...ethnic media..." what is ethnic media?

Authors’ Comment: A clarification has been made in Lines 309-311.

“Intensive child health promotion and education campaigns using ethnic media (radio, TV channels, newsletters in the native language of the beneficiaries) and social networking was observed to positively affect parental awareness, knowledge and beliefs about infectious disease prevention in children”.

Line 285: "...navigate the healthcare system...(REFERENCE MISSING)

Authors’ Comment: The reference is 121, sentence moved before citation number [121] Lines 314-315.

This type of health advisor facilitates communication, establishes the sense of community and helps patients navigate the healthcare system.

Line 302: see my previous comment on vulnerable

Authors’ Comment: The sentence has been modified: “For individuals with additional health needs, such as those requiring prenatal of pediatric care, a Culturally Appropriate Resources and Education (C.A.R.E.) Clinic Health Advisor is recommended for specialty clinics”.

The term “vulnerable” has been removed, Line 312.

Line 310: "Maybe quickly summarise findings here" = remove

Authors’ Comment: The sentence has been removed.

Line 354: Please cite the UN SDGs as you make reference to it

Authors’ Comment: The reference has been added in Line 384.

“Transforming our world: the 2030 Agenda for sustainable development. In: Seventieth General Assembly, New York, 25 September 2015. New York: United Nations; 2015 (United Nations General Assembly resolution 70/1; http://www.un.org/ga/search/view_doc.asp?symbol=A/ RES/70/1&Lang=E, accessed 15 April 2020)”.

Line 356: Emphasise migrants and refugees in your conclusion

Authors’ Comment: Emphasis has been made in Line 386.

“These services, as emerged from our scoping review, include health promotion, disease prevention and disease management activities and should aim to meet the needs of all people, especially migrants, refugees and the vulnerable”.

With kind regards,

Elena Riza, MPH, MSc, PhD

Round 2

Reviewer 1 Report

I feel this new version does a better job focusing on the results of the scoping review. I feel there are still scientific limitation in the process of identifying best practices, however this are pragmatic findings. 

Author Response

Our sincere thanks for the favourable review of our revised manuscript. Your comments are valid and are taken into serious consideration by all authors.

Our effort was to provide insight into some successfully implemented community-based health interventions for migrants and refugees. We referred to them as best practices to be in line with the requirements of our European project. Hopefully, these successful interventions will assist the formation of future interventions at the European level that will facilitate adequate healthcare provision for migrants and refugees.